# New Graph-Based and Transformer Deep Learning Models for River Dissolved Oxygen Forecasting

Paulo Alexandre Costa Rocha [1,2,*], Victor Oliveira Santos [1], Jesse Van Griensven Thé [1,3] and Bahram Gharabaghi [1,*]

1  School of Engineering, University of Guelph, 50 Stone Rd E, Guelph, ON N1G 2W1, Canada; volive04@uoguelph.ca (V.O.S.); jessethe@gmail.com (J.V.G.T.)
2  Mechanical Engineering Department, Technology Center, Federal University of Ceará, Fortaleza 60020-181, CE, Brazil
3  Lakes Environmental Research Inc., 170 Columbia St W, Waterloo, ON N2L 3L3, Canada
*  Correspondence: pcostaro@uoguelph.ca (P.A.C.R.); bgharaba@uoguelph.ca (B.G.)

**Abstract:** Dissolved oxygen (DO) is a key indicator of water quality and the health of an aquatic ecosystem. Aspiring to reach a more accurate forecasting approach for DO levels of natural streams, the present work proposes new graph-based and transformer-based deep learning models. The models were trained and validated using a network of real-time hydrometric and water quality monitoring stations for the Credit River Watershed, Ontario, Canada, and the results were compared with both benchmarking and state-of-the-art approaches. The proposed new Graph Neural Network Sample and Aggregate (GNN-SAGE) model was the best-performing approach, reaching coefficient of determination ($R^2$) and root mean squared error (RMSE) values of 97% and 0.34 mg/L, respectively, when compared with benchmarking models. The findings from the Shapley additive explanations (SHAP) indicated that the GNN-SAGE benefited from spatiotemporal information from the surrounding stations, improving the model's results. Furthermore, temperature has been found to be a major input attribute for determining future DO levels. The results established that the proposed GNN-SAGE model outperforms the accuracy of existing models for DO forecasting, with great potential for real-time water quality management in urban watersheds.

**Keywords:** pollution; dissolved oxygen; Credit River; machine learning; graph neural networks; SHAP analysis

## 1. Introduction

In the Anthropocene Era [1], the increasing urbanization of natural environments close to rivers and lakes has negatively affected the health of the aquatic ecosystems [2,3]. Effluents, specifically urban stormwater runoff and combined sewer outflows, serve as significant sources of pollutants that consume the oxygen in the water through chemical and biological oxygen demands, disrupting the physical, chemical, and biological integrity of the aquatic ecosystem [4,5]. Monitoring the river water quality is important to keep track of the river's health condition, helping develop management strategies aimed at the river's preservation and conservation [6]. One major indicator of the river's water quality is the dissolved oxygen (DO) concentration. The DO is highly variable in both time and space within the stream network, under the constant influence of both groundwater discharge and surface runoff to the stream [7,8]. Fluctuations on DO concentrations can be related to adverse thermal impacts of stormwater runoff and eutrophication due to excessive growth of aquatic plants, impairing aquatic habitat [9–12].

Given the major role of DO in maintaining the health of aquatic ecosystems, it is crucial to forecast and monitor its level with high accuracy [13]. The DO levels can be evaluated by different methods, such as chemical analysis, and by the employment of

sensors. While accurate, chemical analysis, like iodometric titration, is difficult to perform and best suited for lab use. Polarography, an electrochemical-based sensor approach, is widely used but requires regular maintenance and cannot continuously measure oxygen. The fluorescence method, a sensor based on the optical properties of DO, offers quick and sensitive measurements without oxygen depletion. Current studies aim to enhance materials in electrochemical and optical sensors. Intelligent sensors now offer real-time, accurate oxygen measurements through smart signal processing [14,15]. Accurate identification of DO levels helps stakeholders' decision-making and managerial actions, mitigating the impacts of anthropogenic changes on the aquatic environment and aiding the development of strategies to prevent further degradation of the environment.

In this context, machine learning, a subfield of artificial intelligence, has emerged as a powerful tool for enhancing DO monitoring and prediction, providing insight into the factors underlying future DO levels. This paradigm makes use of mathematical tools like probability, optimization, linear algebra, vector calculus, analytic geometry, and matrix decomposition, enabling the analysis of large quantities of information, as well as learning intricate relationships in the data [16]. The principle of machine learning is to create an algorithm that can identify the statistical structures that determine the rules of a dataset through the results provided to it by training rather than being explicitly programmed. In recent years, thanks to the progress in hardware development [17,18], this approach has drawn attention from researchers due to its improved computational performance and generalization capabilities when compared with numerical models. It can also identify complex non-linear associations in the dataset without predefined algorithms to specifically perform this task [19,20]. Despite their attested good performance, ML models may suffer from low data quality, availability, and overfitting. They are also a black-box approach, which hinders the interpretability of their results.

When used in time-series forecasting applications, the ML paradigm reached excellent results in different knowledge fields [21–24], also showing improvements when using past information, i.e., time lags [25–27]. Extensive literature can be found for hydrological studies using ML tools, including studies of water quality and DO. A plethora of studies have shown that different ML and hybrid ML models can forecast DO levels in distinct locations [28–30]. In the context of urban rivers, the ML paradigm also proved to be a valuable asset in DO estimation [6,31–33].

Deep learning (DL) models represent an advanced approach that addresses the limitations of traditional methods in predicting future DO concentrations. Dissolved oxygen estimation is a spatiotemporal problem where recurrent neural networks and their variations are popular among researchers in this field. In [34], the authors used remote sensing data to estimate future DO concentrations using prior DO information in the Rawal watershed in Pakistan. The best result for future DO estimation was returned by the bi-directional long short-term memory (Bi-LSTM) paradigm, with an RMSE equal to 0.199 mg/L. Hybrid configuration merging the LSTM and the convolutional neural network (CNN) paradigm was evaluated in [35] to forecast short-term DO in Small Prespa Lake, Greece. Compared with the standalone models and conventional ML paradigms such as decision trees (DTs) and support vector machine (SVM), the hybrid configuration surpassed all of them, reaching an RMSE of 0.518 mg/L. Other application using CEEMDAN together with DL models was proposed in [36]. In their paper, the authors compared several DL models with real-time forecasting of DO concentrations in Xin'anjiang River, China. They compared the standalone DL approaches CNN, LSTM, the hybrid CNN–LSTM, and combinations of these models with the CEEMDAN pre-processing. The authors discovered that the CEEMDAN–CNN–LSTM hybrid configuration was able to provide the best outcomes for different multi-step ahead forecasting for DO, with the best RMSE value of 0.26 mg/L for 4 h ahead forecasting. Application of DL models in an urban river was investigated in [37]. In their work, the authors implemented a recurrent neural network to forecast DO in Fanno Creek in Oregon, USA, achieving excellent results for DO estimation for 1, 3, and 7 days ahead. In recent years, graph neural network (GNN) models have been implemented to time-series

forecasting within the DL field, reaching cutting-edge results [25,26,38]. However, little is known about implementing such a paradigm in the water quality research area, with few works addressing this subject [39–42].

As the present bibliographic review elucidates, ML applications in hydrological and environmental situations constitute a vast field of study. Aiming to deepen the understanding of ML, specifically addressing the graph-based approaches, the present work aims to contribute to filling the knowledge gap regarding these models when applied to DO forecasting. To this end, historical data comprising the years 2016 to 2020 for Credit River, Ontario, were collected and implemented to the Deep Neural Network Transformer (DNN-Transformer) and the Graph Neural Network Sample and Aggregate (GNN-SAGE) to determine future levels of DO on the river, a highly non-conservative substance. The present study aims to contribute to the field by

1. Developing a cutting-edge model to predict DO concentration with elevated precision and accuracy.
2. Verifying the temporal effect over the estimated DO concentration, a highly non-conservative substance, by implementing different time lags for different predictive models, namely XGBoost, DNN-Transformer, and the proposed GNN-SAGE.
3. Conducting a Shapley additive explanation (SHAP) analysis to assess the significance of different input variables, allowing meaningful inputs over the models' forecasting and its functioning.
4. Enabling the development of a water quality forecasting system for urban rivers, aiding the elaboration of risk management strategies and environmental policies.

## 2. Case Study

The Credit River is a major river in Southern Ontario, Canada, part of the Toronto Greater Area [43] (Figure 1). Starting at Orangeville, the Credit River stretches for 90 km before finally reaching Lake Ontario [44,45]. The river has a mainly rural configuration in its upper and middle areas, having significant forest coverage [46]. As it approaches Lake Ontario, the urbanization of the river's area increases [6]. Urban development in the river area, however, endangers aquatic life and the people who rely on the river, increasing the risk of floods [44,47–49]. Table 1 compiles the Credit River's watershed characteristics. Figure 1 shows the location of the studied site in Ontario province.

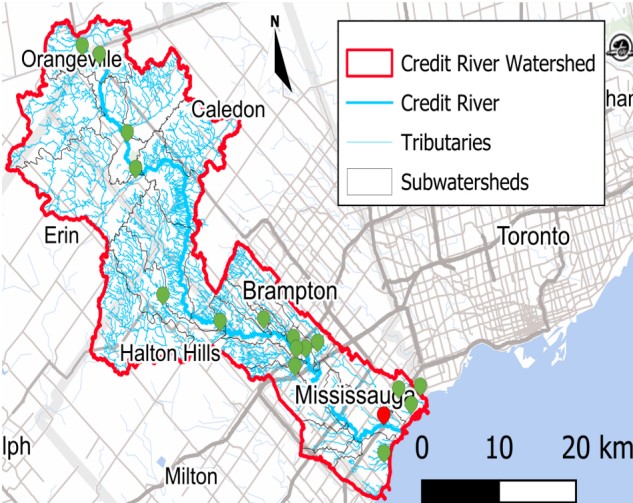

**Figure 1.** Location of the Credit River in Ontario Province. The points showcase the position of the measurement stations along the river's course. The red point in the map represents the study station where the dissolved oxygen is predicted, namely "Credit River @ MGCC", i.e., Mississauga Golf and Country Club. The green points are the used neighboring monitoring stations, which provide spatiotemporal information to the forecasting model.

**Table 1.** Characteristics of the Credit River watershed [43–46,50].

| Credit River Watershed Characteristics | |
|---|---|
| Drainage area | 93,000 ha |
| Credit River length | 90 km |
| Altitude | 190–521 m |
| Area used for agriculture | 35% |
| Area used for urban settlement | 27% |
| Area of natural habitats | 38% |
| Estimated population within the watershed area | 1 million people |

This study developed the forecasting models using historical data from 2016 to 2020. The data were made available by the Credit Valley Conservation Organization (data available at https://cvc.ca/real-timemonitoring/, accessed on 1 November 2023). The monitored data include weather attributes and physical–chemical information collected from stations throughout the river's watershed (Figure 1). The dataset, originally with a temporal resolution of 15 min, was resampled to 1 h intervals using the moving average technique. This adjustment involved calculating the average value for each variable, except for precipitation, which was summed. These processed data were then used as input for the models under evaluation [6,50]. Table 2 shows the statistical description for DO data for all the monitoring stations and the reference station alone, in mg/L. Figure 2 shows the measured DO levels at the reference station "Credit River @ MGCC", for different time windows. After that, the data correlogram is presented in Figure 3, where the target attribute is the dissolved oxygen (first row and first column).

**Table 2.** Statistical description for DO levels recorded by all the monitoring stations and the reference station. The values are in mg/L.

| | All Monitoring Stations | Study Station Credit River @ MGCC |
|---|---|---|
| Minimum | 0.9 | 5.6 |
| Maximum | 16.7 | 16.7 |
| Mean | 10.3 | 10.8 |
| Standard deviation | 2.2 | 2.2 |
| 25% Quantile | 8.9 | 8.9 |
| 50% Quantile (median) | 10.4 | 11.0 |
| 75% Quantile | 12.0 | 12.7 |

In Figure 2, the variance in the DO levels is not visibly meaningful. Panel (a) shows yearly variation in DO for the whole dataset, elucidating that the measured levels' peaks and valleys are well comprised within the maximum and minimum values described in Table 2. Figure 2b elucidates that, for an annual measurement of DO, the temperature effect dominates the behavior of the oxygen levels in the watershed, decreasing during the hotter months of the year and returning to increase as the climate cools down. In Figure 2c, we observe that the temperature still dominates the behavior of DO oscillations, albeit in a time-scale variation ranging from 5 to 7 days. For the daily variance of DO levels presented in Figure 2d, the solar irradiance now dominates the variations on the dissolved oxygen. The data presented in Figure 2 allow us to conclude that the reference station complies with the Canadian Council of Ministers of the Environment (CCME) for DO levels in the freshwater for conservation of the aquatic life [51], indicating that the efforts on maintaining the aquatic ecosystem are yielding positive results.

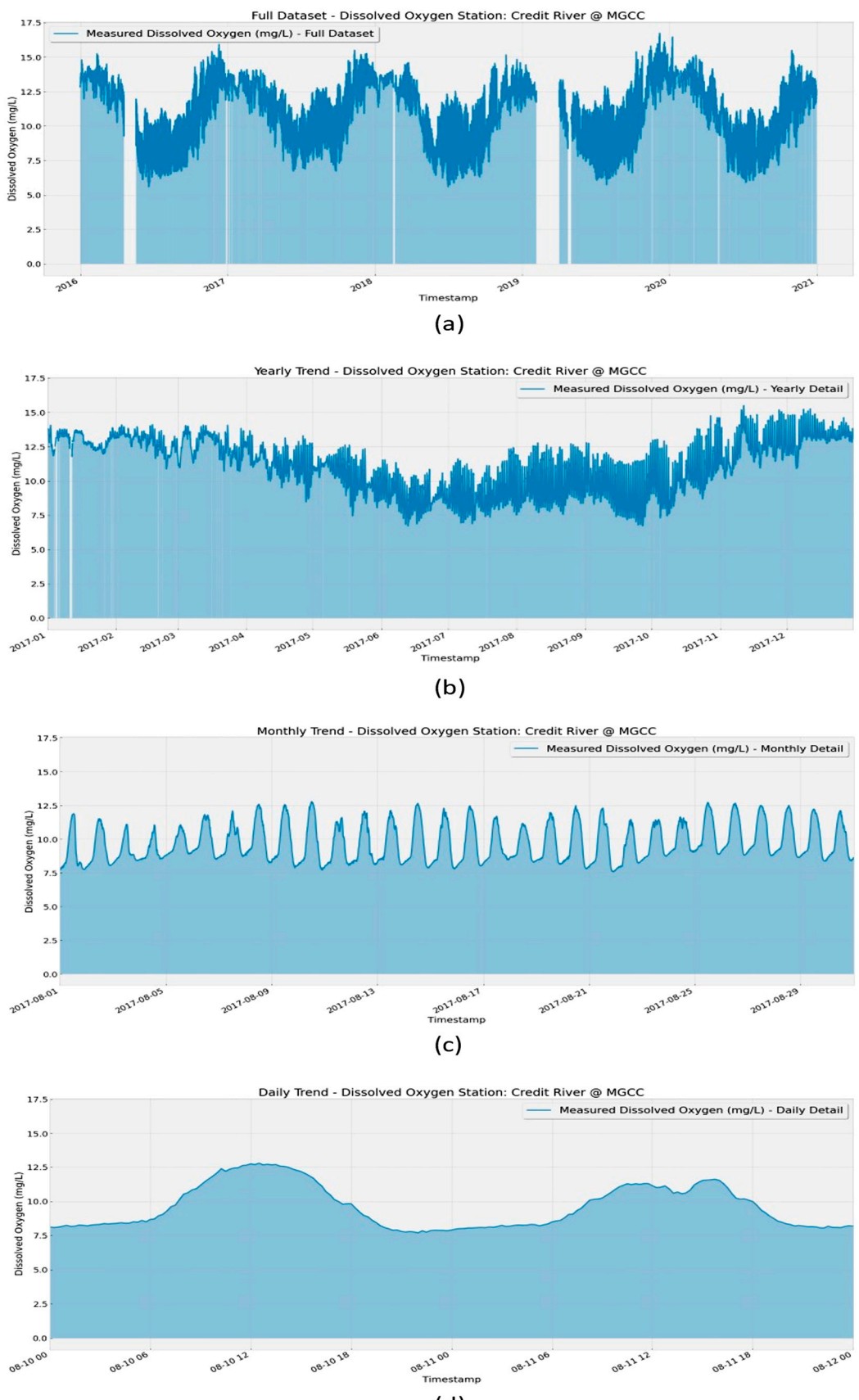

**Figure 2.** Dissolved oxygen measurement for (**a**) the whole dataset, (**b**) one year, (**c**) one month, and (**d**) two sequential days at the reference station "Credit River @ MGCC".

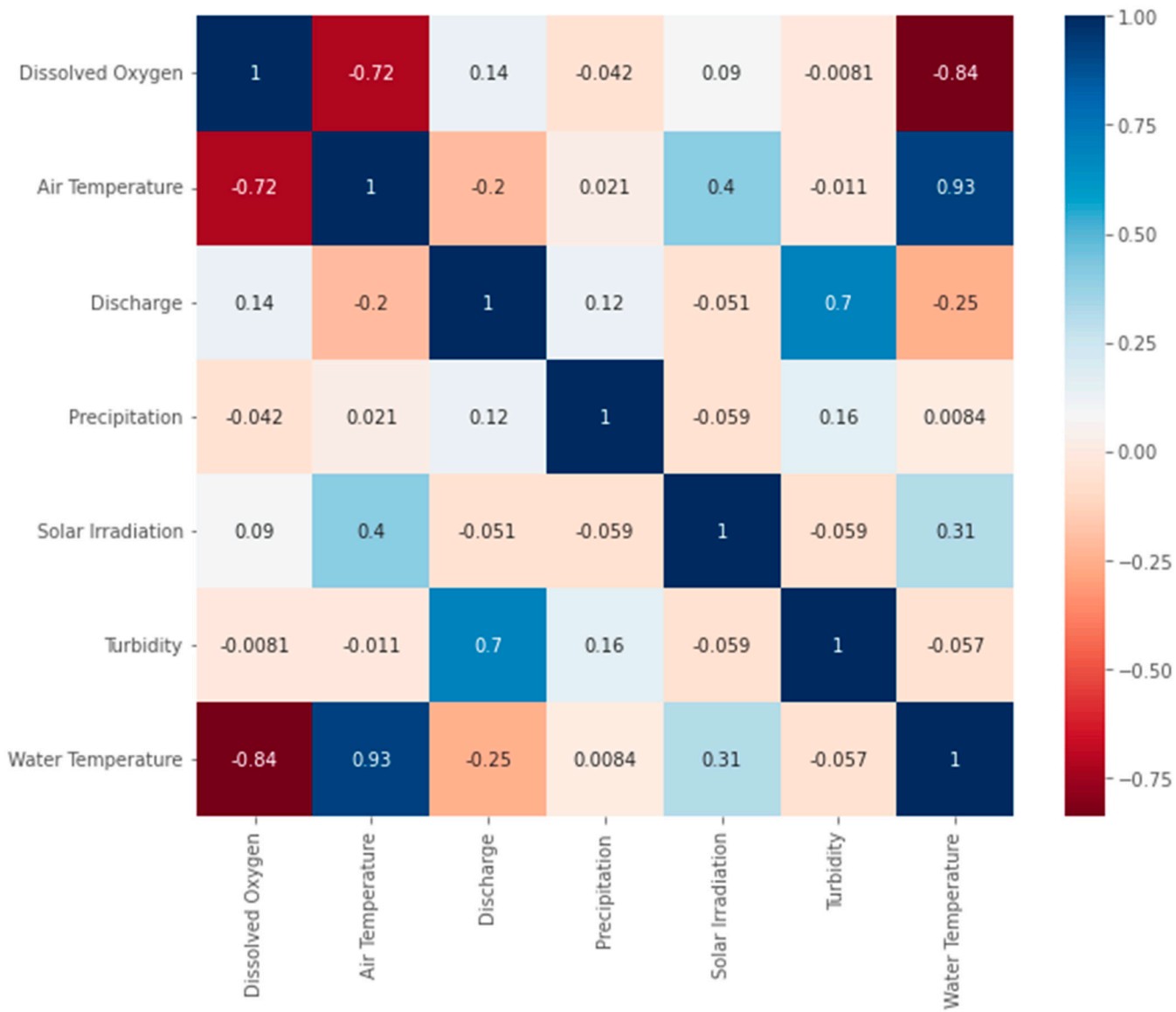

**Figure 3.** Correlation matrix for the Credit River variables used for dissolved oxygen concentration forecasting using Pearson correlation coefficient [52]. The blue colors indicate positive correlation, and the red colors show negative correlation.

In Figure 3, the used input data present a strong negative correlation between water temperature and air temperature, an expected behavior for DO since higher temperatures impair gas solubility in water [53–55]. We observe that positive correlation occurs between the discharge variable and DO due to the discharge's influence on the disturbance to the water body, leading to oxygen transport throughout the river's course. Similar behavior can be noticed between DO and solar irradiation, given the occurrence of the photosynthesis, which also increases oxygen levels on the river. The solar irradiance, albeit responsible to define daily cycles within the river, has its effect as not immediate in this sense. In fact, the exposure to sun light will affect the temperature in the river, which is related to the DO levels in the watershed. This can be visualized in Figure 3, where the correlation between solar irradiance and water temperature (0.31) is similar to that between solar irradiance and air temperature (0.4), indicating a positive correlation between them.

The remaining precipitation and turbidity showed a very small correlation with the dissolved oxygen attribute, indicating that they may not directly affect the DO levels in the

water body. Also, the negative value of these variables indicates that they may reduce the photosynthetic process by the aquatic flora [56,57].

It is important to note that the occurrence of high correlation values in different input attributes does not necessarily positively impact the forecasting result. This may introduce high variance into the predictive model due to collinearity, sometimes damaging its performance [58,59]. However, incorporating input variables with low correlation values into the forecasting model may be advantageous. These inputs may contain valuable spatiotemporal information that can help reveal the connection between the input and output parameters, ultimately improving the model's predictions [50].

## 3. Methodology

### 3.1. Benchmarking Models

The achievements of the assessed models were evaluated against the results of two benchmarking models: the persistence and eXtreme Gradient Boosting (XGBoost). The persistence model is a simple benchmarking method widely used to evaluate any forecasting model. It simply states that the current value of the target variable is the same as the previously measured one. This approach has good performance on forecasting values in short time windows, but as the forecasting horizon increases, it can no longer capture the complex non-linear behavior underneath the dataset nor external factors influencing future values, quickly deteriorating its results [60,61].

XGBoost is a tree-based ML model, improving the random forests approach. Based on the bagging sampling, the XGBoost algorithm generates smaller tree models. It later combines them into a unique bigger one, reducing the final variance and, therefore, the risk of overfitting [62–64]. The XGBoost model performs remarkably well in handling missing values in the dataset, and by being scalable to different applications [65], it is successfully applied in various research fields [66–68].

### 3.2. DNN-Transformer Model

The evaluated DNN-Transformer model utilizes a deep learning technique that incorporates a self-attention mechanism. This enables the model to concentrate on important features of the input information, improving its decision-making capabilities [69]. On its original configuration [70], the transformer architecture determines how the data attend to each one of its elements by implementing positional encoding together with multi-head attention, later being processed by a feed-forward structure that will output the processed data by the encoder. The generated information from the encoder is later used in the decoder structure, where a multi-head structure will again process it before being fed into another feed-forward internal model, which will output the transformer result.

For the present work, the transformer architecture was adapted so it can be applied to a time-series forecasting model. The DNN-Transformer architecture used in the present study is depicted in Figure 4.

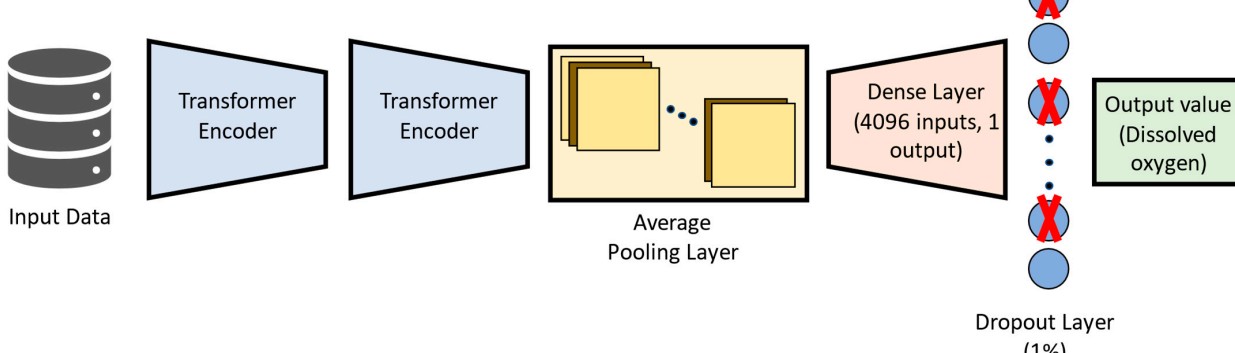

**Figure 4.** Applied DNN-Transformer model structure.

In Figure 4, the input data pass through two transformer structures. Then, their output is processed by an average pooling layer, which connects to a dense layer. The dropout layer [71] is then used to avoid overfitting by relying on an ensemble of different models after the training phase, thus reducing the model's variance [63,72]. Finally, the forecasted value for dissolved oxygen is output by the DNN-Transformer model.

### 3.3. GNN-SAGE Model

The Graph Neural Network Sample and Aggregate (GNN-SAGE) is a cutting-edge model that applies both graph theory and deep learning techniques. Unlike traditional ML models, the graph neural network approach naturally considers graph-structured data, retaining relevant spatiotemporal information ruling the behavior of input variables and target variables, boosting time-series forecasting models [73,74]. For the SAGE paradigm, the model can generalize unknown data by sampling a constant set of nodes, later aggregating them using an aggregator function [75–77], achieving excellence in extracting spatiotemporal information underneath the data [25,38]. The structure used in the present work is depicted in Figure 5.

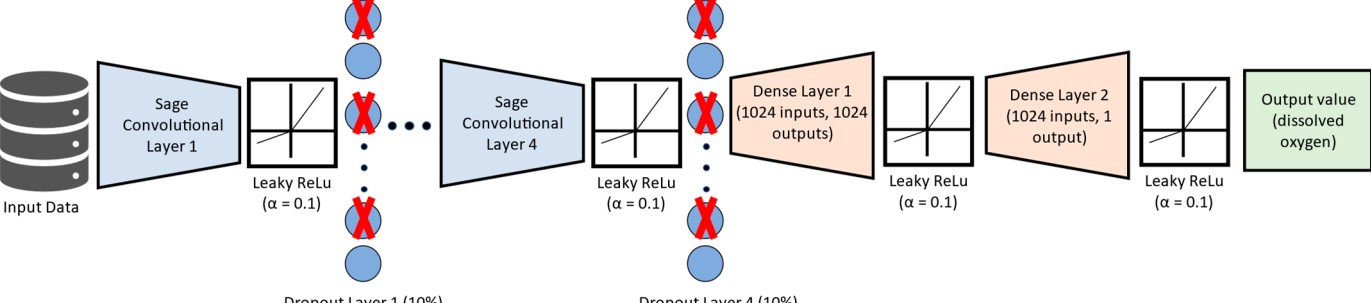

**Figure 5.** Applied GNN-SAGE model structure.

Figure 5 feeds the input spatiotemporal data into the GNN-SAGE model. First, they pass through a succession of graph convolutional layers, followed by the Leaky ReLU function [78,79] and a subsequent dropout layer. After four convolution processes, the data are fed to a dense layer containing 1024 inputs and 1024 outputs, followed again by the Leaky ReLU activation function and a final dense layer containing 1024 inputs and 1 output. Finally, future dissolved oxygen values are then generated by GNN-SAGE.

One key advantage of graph-based architectures, like GNN-SAGE, over traditional DL approaches is their natural ability to process multi-spatiotemporal data from surrounding monitoring stations. This enables them to effectively uncover the inherent connection among input and target variables when implemented in predictive applications [50,73,74]. The capacity of GNN-SAGE to extract spatiotemporal information from the data is relevant to this study, given that DO levels depend on the dataset's temporal and spatial features.

Each model was developed using spatiotemporal data from the monitoring stations and past information called time lags. In this work, the time lag consists of a subseries of the original time series dataset, representing past information in hours, that is fed to the model. That is, being $x_i$ the attribute value taken in the i-th hour, its time lag is defined as a subgroup of the attribute x, where $\{x_{i-1}, x_{i-2}, \ldots, x_{i-n}\}$ contains up to n time lags. The time lag in this study has temporal resolution of 1 h to match the resampling mentioned in the previous section.

For training and testing phases, the dataset comprising the years 2016 to 2020 was split into the ratio 80/20. In this case, the data from 2016 to 2019 were used for training the XGBoost, DNN-Transformer, and GNN-SAGE models, while the remaining year 2020 was reserved for testing these models for a forecasting horizon of 6 h.

### 3.4. SHAP Analysis

Despite the excellent results achieved by ML and DL models in recent studies, these models still lack interpretability, posing as a major hindrance to a full understanding and interpretation of how they work [80–82]. Aiming to solve this problem, the Shapley additive explanation (SHAP) statistic was developed to be a valuable tool for better understanding the black-box behavior of ML applications. This game-theory-based technique provides a deeper understanding of ML models by examining the relationship among input attributes and the resulting outcome. It does so by combining additive feature attribution in terms of each input variable's significance, correlation, and impact on the final prediction, resulting in a better understanding of how complex models work [83].

### 3.5. Evaluation Metrics

The studied models were evaluated using different error metrics, commonly found in the ML field for time-series applications. Error metrics are an important indicator about the ML models' performance and accuracy, allowing posterior comparison with their peers [84]. The used metrics were RMSE, nRMSE, MAE, nMAE, MAPE, MBE, forecast skill, and $R^2$ [85,86].

## 4. Results

### 4.1. Evaluation of Different Time Lags over the Model's Performance

The performances of the DNN-Transformer and GNN-SAGE approaches were compared with the benchmarking ones for a different number of time lags. In this part, each model was evaluated regarding the RMSE metric. The results are presented in Figure 6 for the training set, analyzing time lags ranging from 1 h to 96 h of previous information.

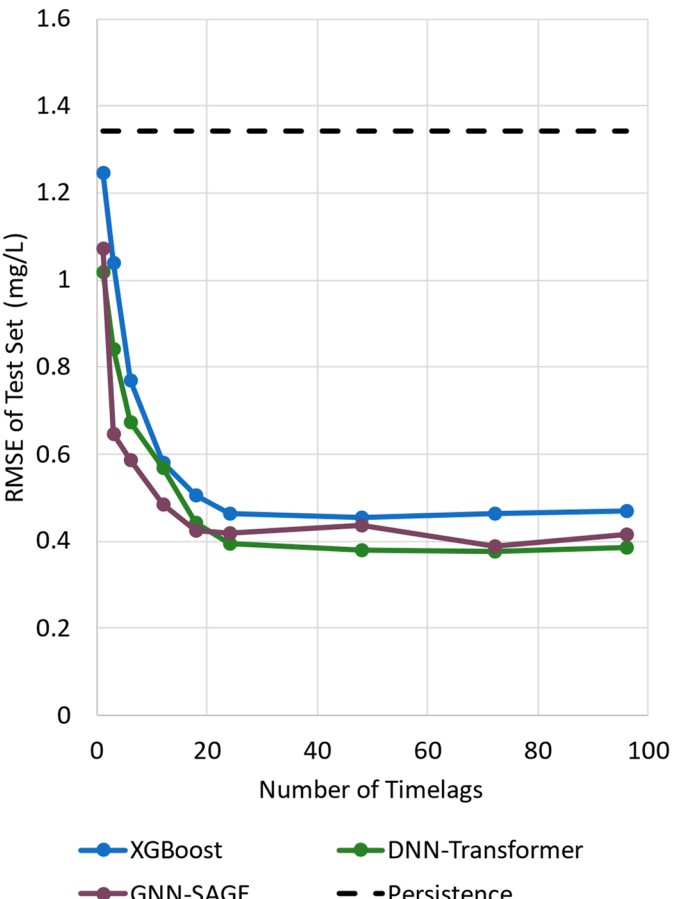

**Figure 6.** Impact of different number of time lags for dissolved oxygen forecasting.

Figure 6 illustrates the evaluated models' performance using different time lags by using only dissolved oxygen as the input parameter. All the models can surpass the persistence and XGBoost benchmarking for the assessed time lag values, presenting similar behavior: a steep decrease in the RMSE value, followed by convergence after 20 time lags. The deep learning paradigms, DNN-Transformers, and GNN-SAGE benefited more as time lags increased. Both XGBoost and DNN-Transformers presented more stable outcomes when compared with GNN-SAGE, which demonstrated a small variation in its results.

For the evaluated models, the best results were achieved by DNN-Transformers for 48 time lags (RMSE = 0.38 mg/L). The proposed GNN-SAGE approach was the second best-performing model, reaching the optimal RMSE value of 0.39 for 72 time lags, a comparable value for DNN-Transformer. The DNN-Transformer model improved dissolved oxygen forecasting by 72% compared with the persistence model and by 17% in comparison with XGBoost. For GNN-SAGE, the improvements were 71% and 15%, when compared with the persistence and XGBoost models, respectively. Finally, using the results depicted in Figure 6, we implemented 72 time lags for the input attributes, i.e., data from up to the previous 72 h for the variables displayed in Figure 3, including dissolved oxygen concentrations, to determine future DO levels using GNN-SAGE, while DNN-Transformer was implemented using 48 time lags.

## 4.2. Results of Dissolved Oxygen for 6 h Ahead

To determine the optimal configuration for forecasting dissolved oxygen levels, various input variables were tested. Using a step-by-step approach [50] for the considered parameters, the models' results were initially evaluated using just past dissolved oxygen information. Additional input parameters were then added to the models, and the resulting forecasts were analyzed. If the addition of an attribute did not improve the model's forecasting performance, leading to a greater error than previous configurations, that attribute was removed. This process was repeated for all models until all attributes depicted in Figure 3 were assessed, considering a 6 h forecasting horizon. The results are presented in Figure 7.

Figure 7 presents the different forecasting performances for each evaluated model. In Figure 7a, it is possible to observe that the XGBoost approach reached improved predictive results as more input variables were added. Its best configuration was set as using dissolved oxygen, air temperature, precipitation, and water temperature, achieving an RMSE value of 0.37 mg/L. Interestingly, the DL transformer-based model's best configuration reached the same error value as XGBoost. Analyzing Figure 7b, using only dissolved oxygen as an input attribute, DNN-Transformer achieved an RMSE outcome of 0.37 mg/L. The model did not need to use any exogenous variables to achieve its best result. Contrarywise, the last panel indicates that GNN-SAGE also benefited from increased input parameters. In fact, it was the best-performing model, reaching an RMSE of 0.34 mg/L when fed with dissolved oxygen, air temperature, precipitation, and water temperature. Interestingly, despite the high correlation depicted in Figure 3 between air and water temperature, the further addition of the latter proved beneficial for the XGBoost and GNN-SAGE models alike, increasing their performance by a tiny margin. The graph-based model enhanced its DO forecasting performance by 8% when compared with both XGBoost and DNN-Transformer.

The models XGBoost, DNN-Transformer, and GNN-SAGE can be evaluated by analyzing Figures 8–10, which contain a scatter plot and marginal distributions for the real measured dissolved oxygen and the predicted values by the models, in mg/L.

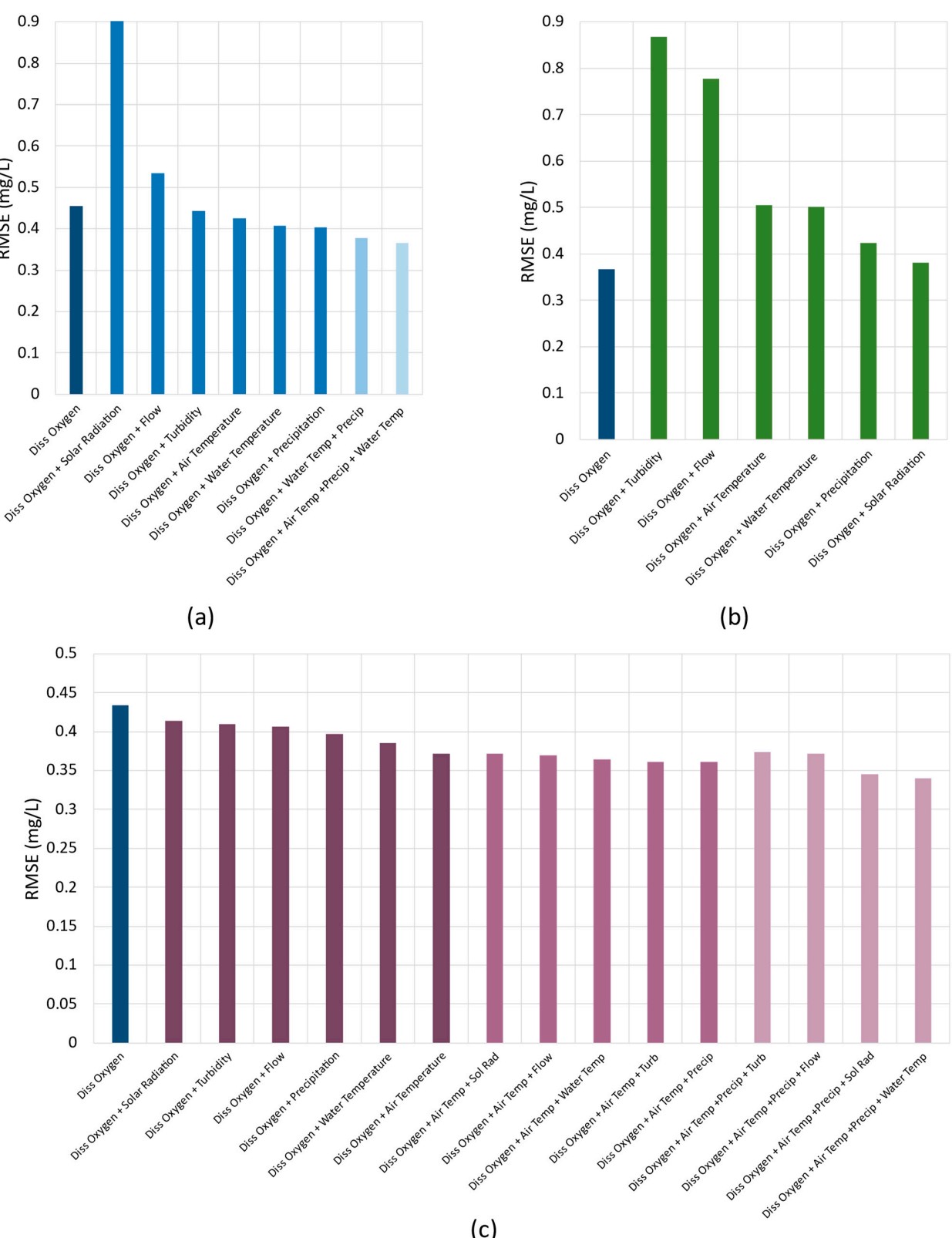

**Figure 7.** This figure depicts the results for this approach, where the *x*-axis shows the used input variables combination and, in the *y*-axis, their respective RMSE values in mg/L. In Figure 7, the lighter the column color, the lower the RMSE value. The effect of different input parameters for dissolved oxygen 6 h ahead estimation using (**a**) XGBoost, (**b**) DNN-Transformer, and (**c**) GNN-SAGE models.

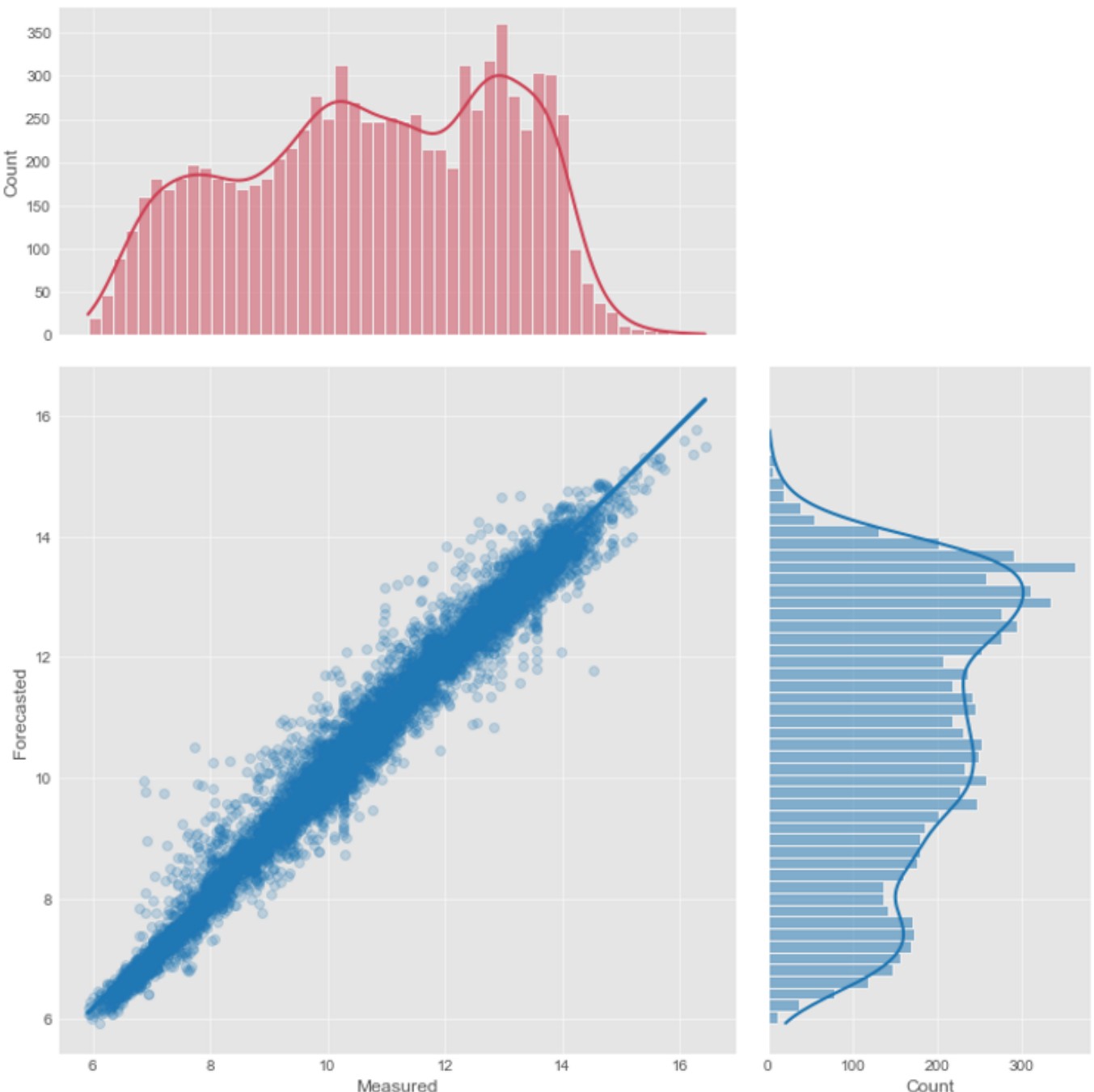

**Figure 8.** Scatter plot illustrating the estimated and observed dissolved oxygen levels for a forecast window of 6 h using XGBoost.

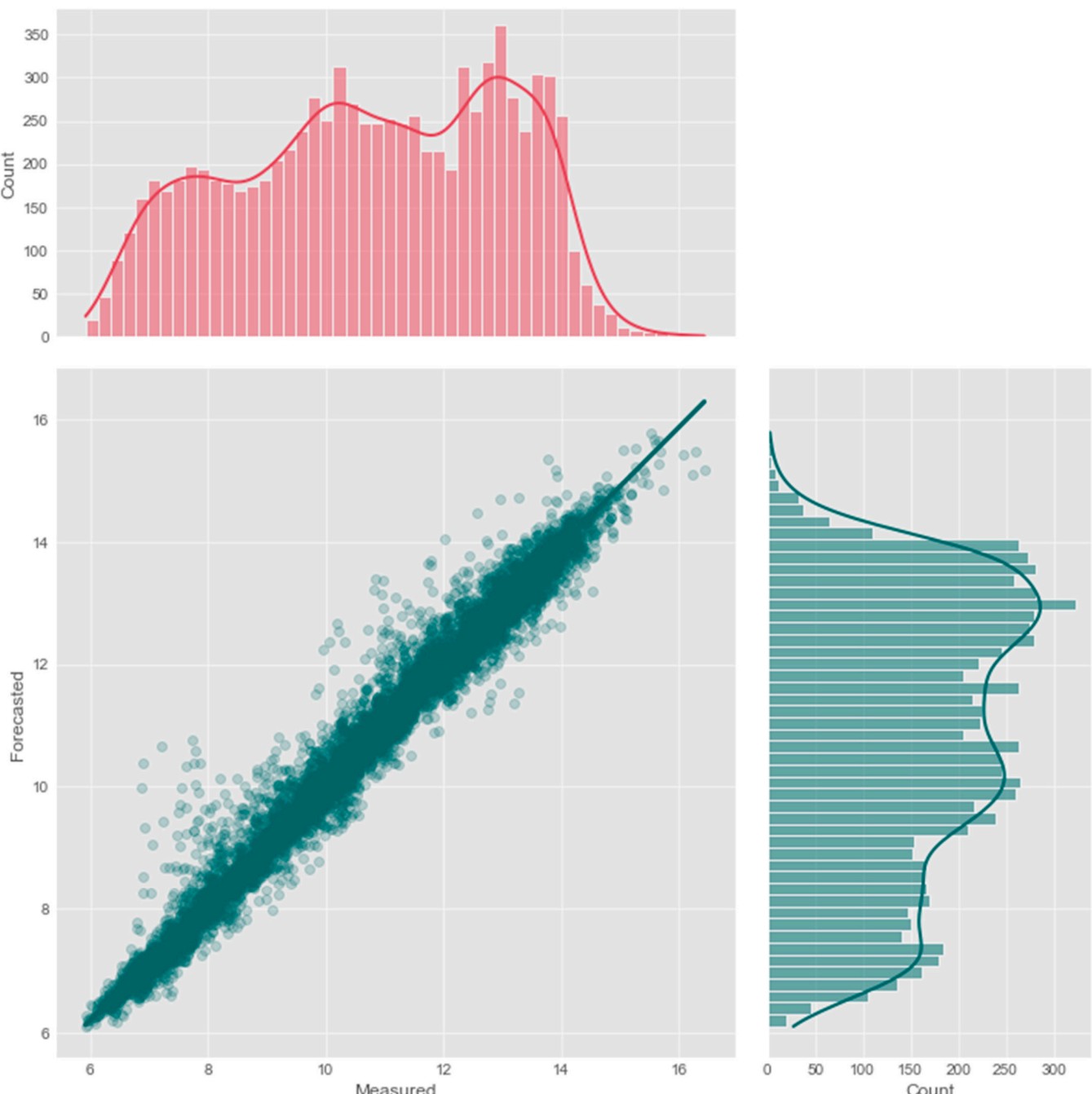

**Figure 9.** Scatter plot illustrating the estimated and observed dissolved oxygen levels for a forecast window of 6 h using DNN-Transformer.

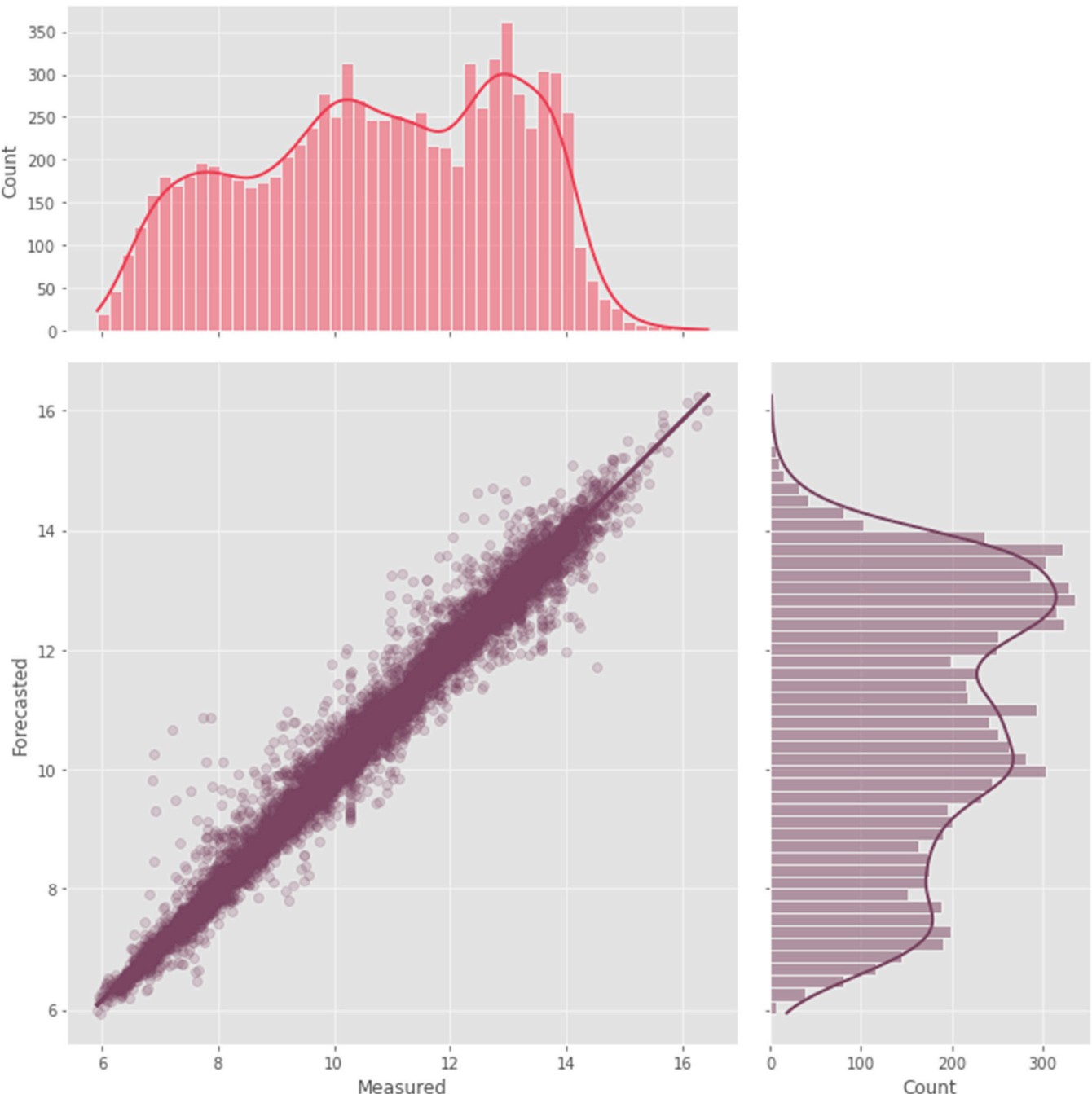

**Figure 10.** Scatter plot illustrating the estimated and observed dissolved oxygen levels for a forecast window of 6 h using GNN-SAGE.

Figures 8–10 present very similar results for the three evaluated models. They have well-aligned scatter points with the regression line, and very similar data distribution, as indicated by the histograms. However, the GNN-SAGE model has superior performance over the tree-based and transformer-based approaches, reaching an excellent coefficient of determination of 97.6%. Figures 11–13 show the model's results over 30 days, displayed on the *x*-axis, comparing them with real measured data for dissolved oxygen in mg/L, displayed on the *y*-axis.

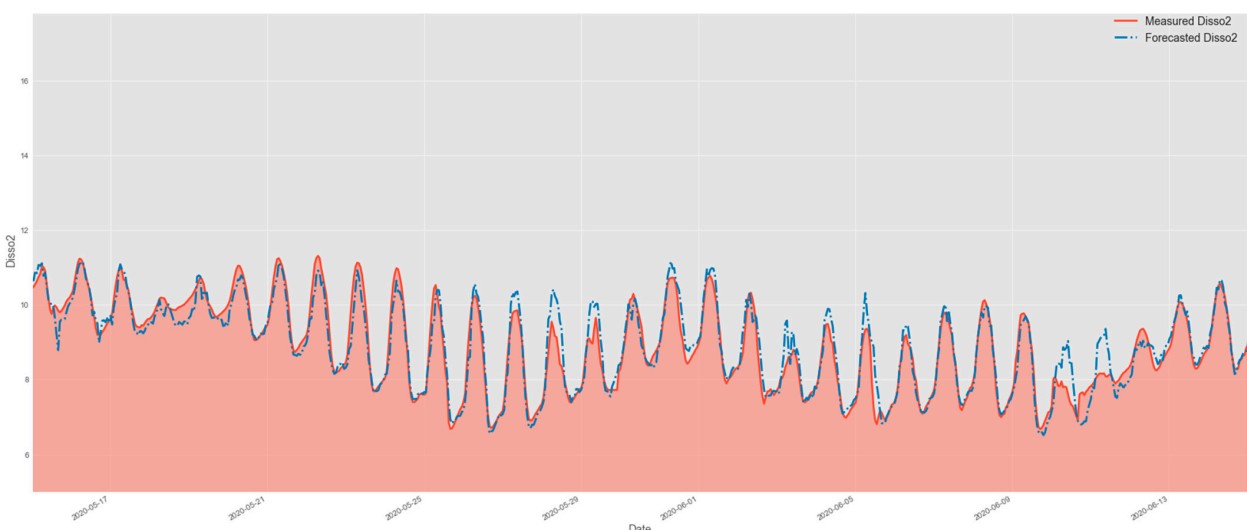

**Figure 11.** Comparison of test data and forecasted data by XGBoost model.

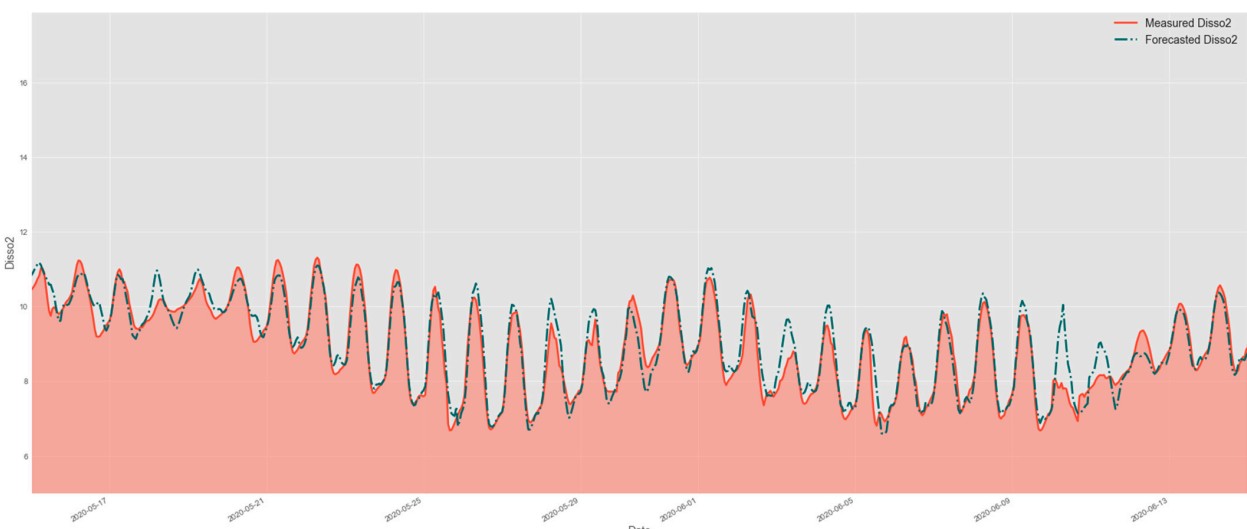

**Figure 12.** Comparison of test data and forecasted data by DNN-Transformer model.

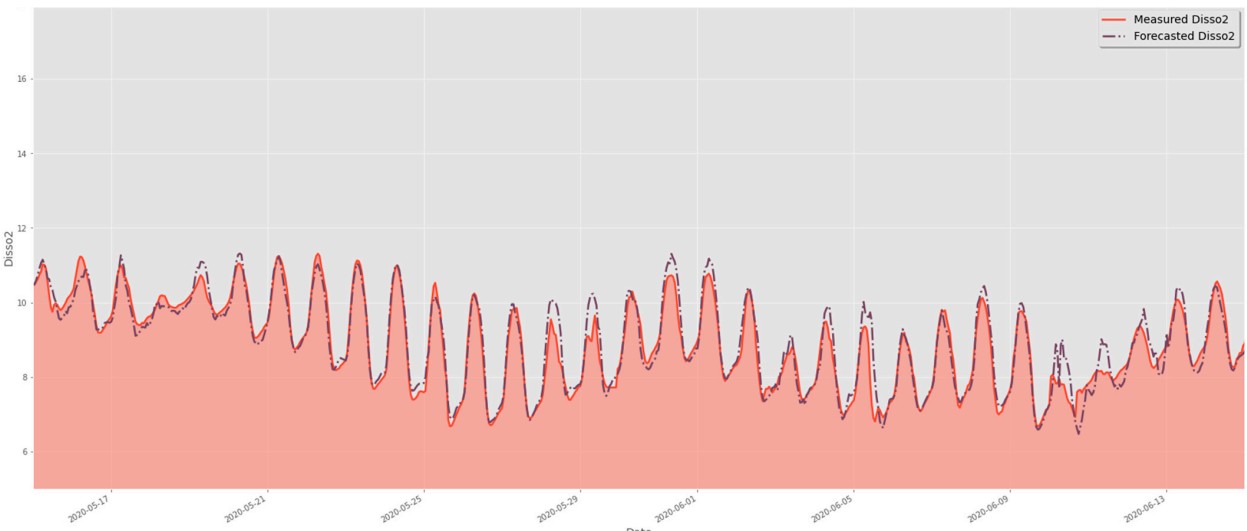

**Figure 13.** Comparison of test data and forecasted data by GNN-SAGE model.

In Figures 11–13, comparing the evaluated models' results for the same assessed period, we can observe that the graph-based structure has a superior performance in identifying the peak concentrations of dissolved oxygen, without any lagging in its results. Comparing XGBoost with GNN-SAGE, we notice that the graph-based approach has superior performance in identifying DO peaks, closely following their shape with a marginal overestimation. Comparing the transformer-based and GNN-SAGE models, we can notice that the DNN-Transformer model tends to maintain the historical behavior of the DO, as seen in the fifth peak from right to left in Figure 12. This is an expected behavior from the transformer approach since it determines the most likely outcome based on the previous sequential data [70]. Contrariwise, GNN-SAGE can aggregate the influence of the spatiotemporal relationships of the input parameters, deeming it sensitive to changes in the DO concentration and thus identifying the peaks within the assessed period, as seen in the same fifth peak from right to left in Figure 13.

## 5. Discussion of the Results

### 5.1. Analysis of the Results of Dissolved Oxygen for 6 h Ahead

When applied to the proposed scenario, the graph-based model GNN-SAGE presented the best performance, providing cutting-edge results for DO forecasting, a highly non-conservative substance, for 6 h ahead forecasting. By using dissolved oxygen, air temperature, precipitation, and water temperature as input attributes, the proposed GNN-SAGE model showed that, when compared with XGBoost and DNN-Transformer, the graph structure can better model and analyze the DO levels. GNN-SAGE can satisfactorily extract spatiotemporal data from the study station and its surroundings, substantially improving the DO forecasting results, reaching $R^2$ of 97.6% and RMSE of 0.34 mg/L, representing improved DO forecasting by 8% compared with both XGBoost and DNN-Transformer RMSE. The superior performance of GNN-SAGE for hydrological applications has been attested in previous studies [38,50].

The proposed model also showed better tracking of DO peaks for 4 weeks, as presented in Figures 8–10. In contrast to the transformer-based approach, the GNN-SAGE model proved to be more sensitive to the spatiotemporal attributes, being able to adapt its forecasting to the current DO concentrations, while DNN-Transformer posed a much more rigid paradigm, tending to maintain the data trend for different DO measurements during the same assessed period, leading to a significant difference between the forecasted and real measured DO. The capacity of GNN-SAGE to adapt to the input spatiotemporal information from the monitoring stations reveals its potential to be used as a real-time monitoring tool for water quality forecasting.

Figures 11–13 show that the assessed models tend to slightly overestimate the DO values for a 6 h ahead time window. This behavior is well known in the ML community. It occurs when the forecasting parameter is too far into the future, leading to a disconnection between the input dataset and the output result. This results in average predicted outcomes [25]. Such difference can also be explained by the lack of sufficient spatiotemporal data provided to the forecasting models under scrutiny, resulting in lagged results as observed in Figures 11–13 [38,87–89].

In addition to these factors, discrepancies between predicted and measured DO values can also be caused by abiotic and biotic factors composing the geochemical environment of the river. Salinity, atmospheric pressure, turbulence, and other geochemical parameters can impact the accuracy of the predictions. Similarly, biotic factors, like photosynthesis and respiration rates of the aquatic flora, can also contribute to the observed discrepancies. The inclusion of these parameters as inputs to the GNN-SAGE approach can improve the model's performance and reduce the difference between the forecasted and real DO values.

Forecasted values greater than the actual DO values can prevent proper decision-making by managers responsible for keeping the watershed's water quality in accordance with regulatory requirements, leading to a deficit in DO. In this sense, the proposed GNN-

SAGE model was the best-performing assessed model, returning the least overestimation on DO levels 6 h ahead, achieving the lowest MAPE value of 2.22%.

### 5.2. Analysis of Results of the SHAP Analysis

Figure 14 shows the results of the SHAP analysis for the 6 h forecasting horizon, where the attributes are arranged in a descending manner from the variable with the most influence on the one with the least influence when used to determine future DO concentrations. The rightmost bar is a scale for the feature value based on the correlation between the variable and the forecasted attribute, indicating that an elevated correlation also has an elevated value. SHAP values greater than zero indicate that positive attribute values positively contribute to DO forecasting, while negative values negatively influence the model's prediction.

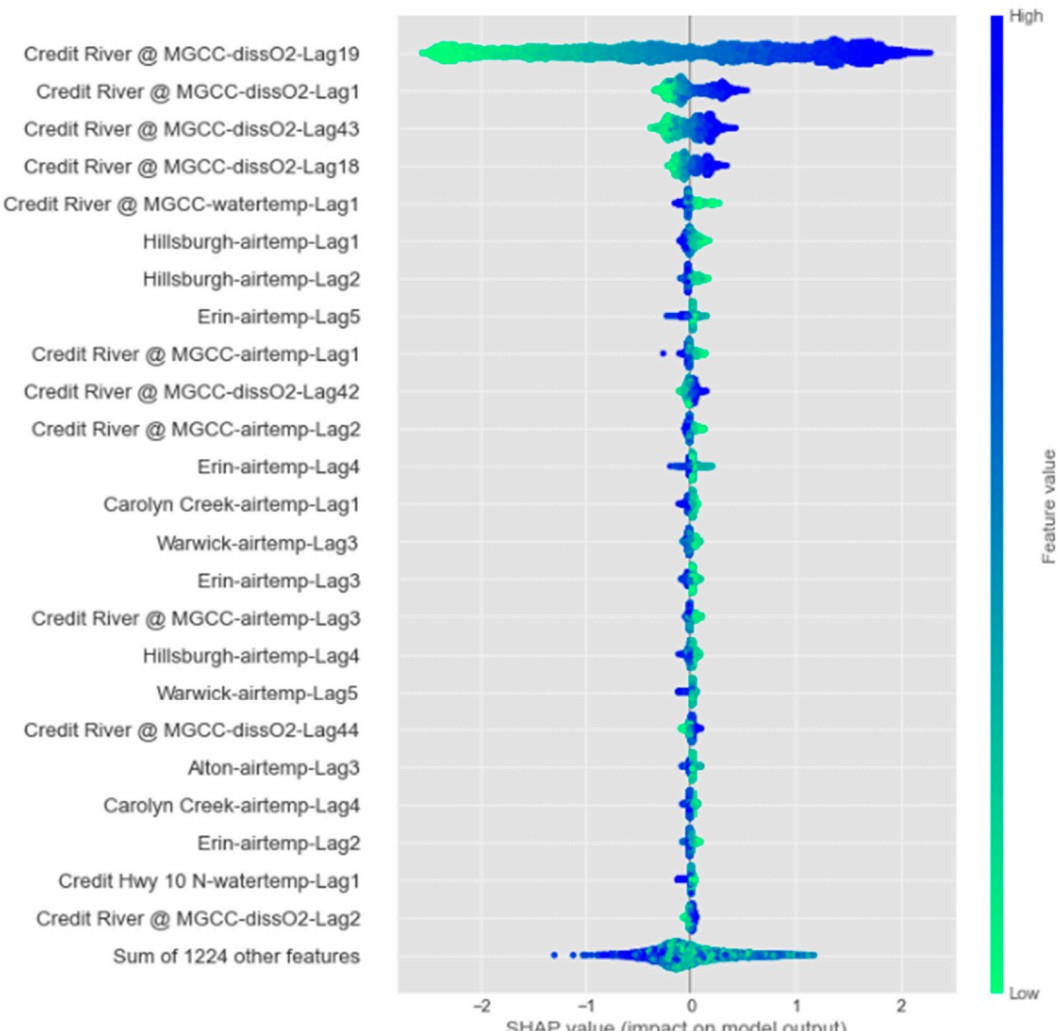

**Figure 14.** Results from the SHAP analysis for predicting DO concentrations 6 h ahead using GNN-SAGE model.

Figure 14 elucidates that using past DO data, i.e., "dissO2-Lag19", "dissO2-Lag1", "dissO2-Lag43", and "dissO2-Lag18", from the reference station "Credit River @ MGCC" provides the most significant amount of spatiotemporal data for the model's forecasting. After that, water and air temperatures are other important factors contributing to the model's DO predictions. For these variables, verifying the relevant influence of spatiotemporal data from the neighboring stations, such as "Hillsburgh" and "Erin", is important, indicating that the model benefits from their information.

The relevant influence of the temperature attribute over the model's result was expected. Figure 3 depicts the elevated correlation between dissolved oxygen and temperature, as elevated temperatures directly affect the oxygen solubility on the water, also contributing to the photosynthesis process and consequent oxygen concentration on the watershed. The importance of the temperature in future estimations of DO has also been studied and verified in previous works [28,42,90], attesting its significant contribution to forecasting models.

The findings from the SHAP analysis provide better insight into the workings of the GNN-SAGE model, showing that it can satisfactorily capture the relationship between the input variables from the study station and its neighboring locations. The spatiotemporal data retrieved from the dataset by the model improved its forecasting capability, with the past DO and temperature data having a relevant influence.

*5.3. Analysis of the Comparison between the GNN-SAGE Results and Literature-Found Values*

In order to understand where the proposed GNN-SAGE approach stands among the current forecasting models, the results yielded by the graph-based model in the current study were compared with the ones found in the literature. Nonetheless, such comparison may not provide a truthful picture between the compared models, meaning that their results are inherent to their specific methodologies, input data, geographic location, and application. Yet, a comparison is still a viable approach to determine how well models can perform when compared with their peers. To facilitate the comparison between the proposed GNN-SAGE with the results from the literature, the following Table 3 compiles the results for different metrics for this model when applied to a 6 h forecasting horizon, and Table 4 presents the results from the literature. For further information on the applied metrics, we recommend reference [85].

**Table 3.** Values of the error metrics for a 6 h forecasting horizon using GNN-SAGE paradigm.

| Metric | Value |
| --- | --- |
| RMSE | 0.34 mg/L |
| nRMSE | 3.17% |
| MAE | 0.23 mg/L |
| nMAE | 2.14% |
| MAPE | 2.22% |
| MBE | 0.01 mg/L |
| Forecast skill | 74.30% |
| $R^2$ | 97.63% |

**Table 4.** Values for dissolved oxygen concentrations forecasting found in the literature.

| Model | Metric Value | Author |
| --- | --- | --- |
| Delft3D | RMSE 1.18 mg/L | Oliveira et al. [7] |
| Delft3D | MAE 1.03 mg/L MAPE 15.9% | Curbani et al. [91] |
| Prophet | RMSE 0.71 mg/L MAE 0.55 mg/L | Kogekar et al. [92] |
| LSSVM-BA | RMSE Mean value 0.79 mg/L MAE Mean value 0.94 mg/L | Yaseen et al. [93] |

**Table 4.** *Cont.*

| Model | Metric Value | Author |
|---|---|---|
| Bi-LSTM | RMSE<br>0.2 mg/L<br>MAE<br>0.15 mg/L | Ahmed et al.<br>[34] |
| CEEMDAN–CNN–LSTM | RMSE<br>0.26 mg/L for 4 h forecasting horizon<br>0.28 mg/L for 8 h forecasting horizon<br>0.31 mg/L for 12 h forecasting horizon<br>0.34 mg/L for 16 h forecasting horizon<br>0.39 mg/L for 20 h forecasting horizon<br>0.48 mg/L for 24 h forecasting horizon<br>(Average RMSE of 0.34 mg/L)<br>MAPE<br>2.55% for 4 h forecasting horizon<br>2.79% for 8 h forecasting horizon<br>3.00% for 12 h forecasting horizon<br>3.30% for 16 h forecasting horizon<br>3.65% for 20 h forecasting horizon<br>4.56% for 24 h forecasting horizon<br>(Average MAPE of 3.31%) | Sha et al.<br>[36] |

In references [7,92], DO was modeled by the physical approach of Delft3D for rivers located in Portugal and Brazil, respectively. From the comparison between the GNN-SAGE results and those using a physics-based approach, it can be concluded that the graph model provides superior results considering RMSE and MAPE, substantially improving DO forecasting. Also, the MAPE comparison allows us to conclude that GNN-SAGE provides more accurate estimates for DO than the model used in [91]. Furthermore, it is worth noting that GNN-SAGE, a graph-based model, can inherently process multi-spatiotemporal information contained in the input dataset, without requiring explicit programming and/or feature preprocessing. Another advantage of the GNN-SAGE paradigm is its simpler implementation: differently from the Delft3D model, which models the DO levels by the finite difference method and thus requires the domain to be defined by a mesh grid, GNN-SAGE uses only the inputs presented in Figure 2.

In reference [92], the authors implemented the Prophet ML model. The Prophet was developed and made available by Facebook's Core Data Science group [94] and is a cumulative approach that excels in analyzing and forecasting non-linear data trends [95]. Their work implemented the Prophet for a DO forecasting task for a river in India. Comparing the GNN-SAGE results with those in [92], the proposed model clearly provides superior results over a simpler ML model implementation such as the Prophet. The superiority of GNN-SAGE is expected in this scenario, once it can process and identify seasonality and spatial information from the dataset, while the Prophet may handle temporal data only. From this comparison, we can conclude that spatial data play a major role in determining future DO levels by incorporating relevant data for the model's prediction, as stated in the SHAP analysis results in Figure 13.

In work [93], the authors proposed a hybrid model called the Least Square Support Vector Machine-Bat Algorithm (LSSVM-BA) for monthly DO estimation in the USA. Their results for the best-performing LSSVM-BA for monthly DO predictions were, on average, 0.79 mg/L and 0.94 mg/L for RMSE and MAE metrics, respectively. When directly comparing these results with those of GNN-SAGE, the graph-based model surpassed the results found in the literature with a significant difference on both metrics, again proving to be a superior approach to this task. In studies [34,36], different DL paradigms were implemented for DO forecasting. Interestingly, among the results found in the literature in Table 2, the DL models were the best-performing ones, offering significant improvement for DO estimation

over both physical and traditional ML approaches, achieving error values within the same order of magnitude. In work [34], Bi-LSTM was implemented to model future DO levels in Pakistan using remote sensing data up to three time steps ahead. The study [36] used a hybrid model consisting of CNN–LSTM and CEEMDAN for predicting DO in China up to 24 h ahead. The CNN–LSTM hybrid configuration is well known for its ability to process spatial and temporal data through its convolutional and recurrent structures, respectively. Both papers reached similar results for DO forecasting regarding the RMSE metric. However, comparison with GNN-SAGE proved that the proposed paradigm in this work can surpass the results for the DL models by a significant margin, providing more accurate and precise DO forecasting.

## 6. Conclusions

In this study, a new GNN-SAGE deep learning model for forecasting DO levels in natural streams was developed using a network of real-time water quality monitoring stations in an urban watershed. The accuracy of the model forecasts for a forecasting horizon of 6 h was compared with the popular, XGBoost and DNN-Transformer models. The best-performing GNN-SAGE configuration was obtained using 72 h time-lag and input parameters dissolved oxygen, air temperature, precipitation, and water temperature from the reference station "Credit River @ MGCC" and its neighboring ones.

The results show that the proposed new GNN-SAGE can gather influence from the provided network of real-time hydrometric and water quality monitoring stations and is able to more accurately forecast changes in DO levels. This leads to an improvement of 8% in the RMSE metric when compared with both XGBoost and DNN-Transformer, and it reaches an $R^2$ value above 97%. The SHAP analysis outcomes elucidated the importance of spatial data coming from the neighboring stations.

By comparing the GNN-SAGE results with those found in the literature, it was possible to attest the superior performance of the graph-based model, which surpassed every assessed model in Table 4 with a significant margin for both RMSE and $R^2$ values. These findings made the GNN-SAGE model a valuable DO forecasting tool, offering precise and cutting-edge predictions in urban river watersheds.

For future applications of the proposed GNN-SAGE model, other hydrological parameters can be assessed using this approach and directly determine the water quality index. Furthermore, more spatial information can be added to future implementations of GNN-SAGE, aiming to reduce its geographical bias once it was modeled and validated for only one location. To this end, spatiotemporal data from different rivers can be added to the model, enhancing its generalization and broadening its geographic application. Future works can also investigate the performance of the proposed GNN-SAGE when applied to estuary regions. Given the significant spatial extent of estuaries, the graph model is anticipated to be highly adaptable for water quality monitoring in these areas. The implementation of a graph-based model to assess water quality on estuaries is a promising topic worth to be further explored.

Developing a precise and accurate model to forecast real-time hydrological parameters is relevant in the current socio-economic–environmental scenario. Technologies like GNN-SAGE have the potential to enhance the creation of new legislation regarding the protection and sustainable use of watersheds, as well as to support stakeholders in their decision-making and risk management strategies. These actions can mitigate further damage from human actions over the aquatic environment when used as a real-time forecasting tool.

**Author Contributions:** Conceptualization, J.V.G.T. and B.G.; methodology, P.A.C.R., J.V.G.T. and B.G.; software, P.A.C.R.; validation, P.A.C.R., J.V.G.T. and B.G.; formal analysis, P.A.C.R.; investigation, P.A.C.R., J.V.G.T. and B.G.; resources, J.V.G.T. and B.G.; data curation, J.V.G.T. and B.G.; writing—original draft preparation, V.O.S. and P.A.C.R.; writing—review and editing, V.O.S., P.A.C.R., J.V.G.T. and B.G.; visualization, V.O.S. and P.A.C.R.; supervision, J.V.G.T. and B.G.; project administration, J.V.G.T. and B.G.; funding acquisition, B.G. and J.V.G.T. All authors have read and agreed to the published version of the manuscript.

**Funding:** This research study was funded by the Natural Sciences and Engineering Research Council of Canada (NSERC) Alliance, grant no. 401643, in association with Lakes Environmental Software Inc., and by the Conselho Nacional de Desenvolvimento Científico e Tecnológico—Brasil (CNPq), grant no. 303585/2022-6.

**Data Availability Statement:** Publicly available datasets were analyzed in this study. This data can be found here: https://cvc.ca/real-timemonitoring/, https://drive.google.com/drive/folders/13Ef-_EklzJze8pZx1oIDoQFKU304d7NF, and https://drive.google.com/drive/folders/136RH-G-nPVScO7Ln7OOC0WEYl3kk5kDW (accessed on 22 September 2023).

**Conflicts of Interest:** The authors declare no conflict of interest.

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
