# Peer review of "New Graph-Based and Transformer Deep Learning Models for River Dissolved Oxygen Forecasting"

_environments, doi:10.3390/environments10120217_

Round 1
Reviewer 1 Report
Comments and Suggestions for Authors
New Graph-Based and Transformers Deep Learning Models for River Dissolved Oxygen Forecasting
Comments
(1) Introduction: The bibliographic review lacks of some references, especially for urban rivers, for example, “Prediction of dissolved oxygen in urban rivers at the Three Gorges Reservoir, China: extreme learning machines (ELM) versus artificial neural network (ANN)”.
(2) Case study: for urban rivers, I am not sure whether other factors will impact DO levels, such as CBOD. It needs investigation because this will impact model input.
(3) I think abbreviations can be used for the variables, such as SR for solar radiation, Q for flow, thus figures will look much better. For example, Fig. 6, the x-label can be revised to: DO, DO+SR, DO+Q…
(4) Figs. 7 to 9 are unclear. The same for Figs. 10 to 12.
Reviewer 2 Report
Comments and Suggestions for Authors
The manuscript submitted by Rocha et al proposes a new model for predicting dissolved oxygen in rivers, based on a deep learning model. The context is the increasing impact of anthropic activities that are likely to have a negative impact on water quality. In this work, dissolved oxygen is retained as a tracer of water quality considering its importance for biota. The study case is the Credit River watershed (Canada). The authors explain the objectives of the work, i.e. to compare the performance of different machine learning approaches in order to demonstrate that their proposed solution, a Graph Neural Network Sample and Aggregate (GNN-SAGE), is the most efficient.
As the manager of a water quality network in a large fluvial-estuarine system affected by seasonal hypoxia, I agreed to read the article with great interest to review. Indeed, a warming system of hypoxia would be very helpful for mitigation. But when I read it, I was disappointed. The article is very technical and quite airtight for a reader unfamiliar with the approaches used. The interest of such an article would be to make this type of technique more popular for the operational management of watercourses and to anticipate quality problems. However, the article presented is too far removed from the reality on the ground to convince potential users. I classify the main drawbacks of the article into main categories:
1/ While the example is based on a real network, there is almost no information on the Credit River watershed. As the authors justify the work by the need to monitor water quality, there is any information on the range of the used parameters and on DO. I have derived DO levels from figures 7 to 9 ! An overview of the spatio-temporal variability of the parameters would have made it possible to understand the strategy underlying the work. For example, there is no explanation of the choice of parameters retained. Are they used because they are available or because the authors had a reflection on those that would be useful.
2/ The rather naive treatment of the data: in addition to the previous remakes (the justification of the selected parameters), the few comments on the data set and the correlation between the parameters are extremely basic (Fig 2) and, in the absence of data examples, sometimes surprising. The low correlation between solar irradiance and DO is surprising as there is often in river a daily signal in DO production. In the absence of data plot, it is not possible to verify. The relationship between air and water temperature is well known in river and these two parameters are not independent, so the use of both is questionable.
The authors explain the weak relationship between discharge and turbidity, “by disturbing the water body” (lines 152-153): what does this mean ? Usually an increase in river discharge is associated with an increase in erosion or sediment resuspension, which is also consistent with the correlation (16%) between precipitation and turbidity. The effet of turbidity and pluviometry on DO is not so simple, it depends on the concentration of SPM but also on the carbon content and the lability of the particle.
3/ Introduction to the different ML models: it is extremely technical with too many references, but a more A more mainstream presentation of their principles and differences is missing. If the authors are convinced that the GNN-SAGE is better, they should promote its use by making it available on a github or similar repository.
The test of the different MLs is not fair. A reliable test uses the same scenario for each model. This is not the case (Figure 6). To be reliable, it is essential that the same parameter combinations are used and compared for each model. This is a major shortcoming of the article.
Again, there is no attempt to relate this to reality. Figure 11 compares the test data (no explanation of the choice of these data) and the predicted DO. Why is this not done for the 3 models ? There are several periods where the two models failed to predict the variation in DO, particularly at the end of the curve (unable to read the X scale). This is not discussed in the article, it would have been very pertinent to discuss these large differences between measured and forecasted, and then to try to understand what happened during this period that could explain why the forecasted values are much higher. Overestimation would be a problem for an operational system designed to prevent DO deficit (e.g. by reducing sewage input). Here again, there is no attempt to link the work to the real world.
Other concerns
What are “residues” (line 33)
Disturbs on DO … are related to eutrophication … (line 41): not only global change also impacts DO (rising temperature, change in pluviometry and river discharge)
Problems of units: whereas the authors refer to published work, they use mg/L and for their work ppm. DO needs to be given always in mg/L
Suppress lines 116 to 119, any interest,
Weather attributes and physical-chemical information (lne 137) to develop which parameters … The map is not readable, increase the watershed area.
Why 1h was selected ? (line 139)
In the water body (line 156) to suppress
Figures 3 and 4: to explain some parameters (dropout later for example) and the associated percentages
Figures 7, 8, 9 to suppress no interest
Leaky ReLu (line 221) to explain
Figure 6: the same scenarios for each test and the same Y scale to permit comparison
Figure 8: to improve, remove the coloured area, increase the text of the title and scales, enlarge the Y scale 6 to 16 would be better to see the differences.
Figure 13 too technical and I do not have a same reading as the authors, for example air temperature is present, not water
Table 2 need to be improved and coherent in term of unit with table 1. Compare for your work and the previous equivalent information. Why all the forecasting horizon are given whereas only the two first are in the equivalent range of the work (6-hour)
In short, the article has the potential to have a wide audience but the current version is too technical. The authors need to produce an article that is useful for other systems, otherwise they should choose a technical journal.
Reviewer 3 Report
Comments and Suggestions for Authors
see attachment

there are a few typos, places with passive voice, and too much jargon, but nothing crazy
Round 2
Reviewer 1 Report
Comments and Suggestions for Authors
I thank the authors by addressing my review comments. I think the current version is acceptable.
Author Response
Dear reviewer, thank so much for your time and consideration reviewing our work. Your insightful comments and suggestions have greatly improved our work, and for that, we are truly grateful.
Best regards,
The authors.
Reviewer 2 Report
Comments and Suggestions for Authors
I've looked at the authors' responses, as the magazine asks for rapid feedback. I note that authors do not make much effort to respond to comments. A table with DO values is not enough (a siglne number after the dot is enoguh !); the trends in this small catchment area need to be described in more detail and, above all, bibliographical references on these data need to be added. If the data had not already been published, it would be even more important to present the dissolved oxygen. "Residus" corresponds to effluent. But this is also too brief, as not all pollutants affect DO .
The article is still very mathematical. I don't agree with the authors' answers on the comparison of the different models. A comparison should be based on the same scenarios, i.e. the same forcing parameters. Similarly, I know that a model may not reproduce the data exactly, but it is critical to understand why there are such discrepancies (see 1st review) for short periods. This is because the system is supposed to make forecasts, and such discrepancies would be problematic. So a discussion of the context and explanatory hypotheses needs to be addressed. I still believe that the 3 figures are not useful. I repeat that authors should choose either a highly specialised journal or the present journal, but should make the effort to interest a wider audience.
The authors do the maths but seem to know nothing about the geochemistry of dissolved oxygen and its behaviour in the environment. In fact, I was interested in the subject because I have a monitoring network in a large estuary and such a possibility of predicting DO would have been of great interest if it had been dealt with properly.
Author Response
Please, see the attachment.

Reviewer 3 Report
Comments and Suggestions for Authors
Thank you for addressing the review comments - the paper has been improved
Author Response

(The authors gave the same response as above.)

Round 3
Reviewer 2 Report
Comments and Suggestions for Authors
the authors persist in making edible modifications. Many sentences and statements show a lack of understanding of dissolved oxygen. For example, the authors suggest that in situ measurement of DO is complicated, which ignores the existence of numerous high-frequency monitoring systems using optical sensors. Yes, DO decreases with increasing temperature, and therefore in warm periods. What about saturation? Given the concentrations finally presented, what is at stake for dissolved oxygen? There are still no answers to the major discrepancies between predicted and measured DO, even though the authors claim to be forecasting. There are already articles on DO prediction (for example, I've just seen a new article on the subject: https://doi.org/10.1016/j.envsoft.2023.105884). For my part, I do not recommend the publication of this manuscript
Author Response
Please, see the attachment.
